# Insights from a comprehensive study of *Trypanosoma cruzi*: A new mitochondrial clade restricted to North and Central America and genetic structure of TcI in the region

Raquel Asunción Lima-Cordón[1]☉*, Sara Helms Cahan[1]☉, Cai McCann[1]‡, Patricia L. Dorn[2]☉, Silvia Andrade Justi[3,4,5]‡, Antonieta Rodas[6]‡, María Carlota Monroy[6]‡, Lori Stevens[1]☉

**1** Department of Biology, University of Vermont, Burlington, Vermont, United States of America, **2** Department of Biological Sciences, Loyola University New Orleans, New Orleans, Louisiana, United States of America, **3** The Walter Reed Biosystematics Unit, Smithsonian Institution Museum Support Center, Suitland, Maryland, United States of America, **4** Entomology Branch, Walter Reed Army Institute of Research, Silver Spring, Maryland, United States of America, **5** Smithsonian Institution–National Museum of Natural History, Department of Entomology, Washington, DC, United States of America, **6** The Applied Entomology and Parasitology Laboratory, Biology School, Pharmacy Faculty, San Carlos University of Guatemala, Guatemala City, Guatemala

☉ These authors contributed equally to this work.
‡ CM, SAJ, AR, and MCM also contributed equally to this work.
* raqueasu7@gmail.com

## Abstract

More than 100 years since the first description of Chagas Disease and with over 29,000 new cases annually due to vector transmission (in 2010), American Trypanosomiasis remains a Neglected Tropical Disease (NTD). This study presents the most comprehensive *Trypanosoma cruzi* sampling in terms of geographic locations and triatomine species analyzed to date and includes both nuclear and mitochondrial genomes. This addresses the gap of information from North and Central America. We incorporate new and previously published DNA sequence data from two mitochondrial genes, Cytochrome oxidase II (COII) and NADH dehydrogenase subunit 1 (ND1). These *T. cruzi* samples were collected over a broad geographic range including 111 parasite DNA samples extracted from triatomines newly collected across North and Central America, all of which were infected with *T. cruzi* in their natural environment. In addition, we present parasite reduced representation (Restriction site Associated DNA markers, RAD-tag) genomic nuclear data combined with the mitochondrial gene sequences for a subset of the triatomines (27 specimens) collected from Guatemala and El Salvador. Our mitochondrial phylogenetic reconstruction revealed two of the major mitochondrial lineages circulating across North and Central America, as well as the first ever mitochondrial data for TcBat from a triatomine collected in Central America. Our data also show that within mtTcIII, North and Central America represent an independent, distinct clade from South America, named here as mtTcIII$_{NA-CA}$, geographically restricted to North and Central America. Lastly, the most frequent lineage detected across North and Central America, mtTcI, was also an independent, distinct clade from South America, noted as mtTcI$_{NA-CA}$. Furthermore, nuclear genome data based on Single Nucleotide Polymorphism (SNP)

**Data Availability Statement:** All relevant data are within the manuscript and its Supporting Information files.

**Funding:** This work was funded by the National Science Foundation (NSF) grant BCS-1216193 (LS, SHC, PD, MCM); NSF grant DGE-1735316 (LS); National Institutes of Health (NIH)-grant R03AI26268/1-2 (LS, SHC); International Development Research Centre (IDRC) grant ID 106531-001 (MCM); University of Vermont (UVM) Graduate college through the Dr. Roberto Fabri Fialho Research Award (RALC) and UVM College of Arts and Sciences Dean (CM). RALC was supported by the Quantitative and Evolutionary STEM Training (QuEST) Program through NSF grant DGE-1735316. This study was conducted while SAJ held a National Research Council Research Associateship at the Walter Reed Biosystematics Unit and Walter Reed Army Institute of Research and was funded in part by the Armed Forces Health Surveillance Division – Global Emerging Infectious Diseases (AFSHD-GEIS) core funding to Walter Reed Biosystematic Unit grant P0030_21_WR. The funders had no role in study design, data collection and analysis, decision to publish, or preparation of the manuscript.

**Competing interests:** The authors have declared that no competing interests exist.

showed genetic structure of lineage TcI from specimens collected in Guatemala and El Salvador supporting the hypothesis that genetic diversity at a local scale has a geographical component. Our multiscale analysis contributes to the understanding of the independent and distinct evolution of *T. cruzi* lineages in North and Central America regions.

## Author summary

Neglected Tropical Diseases (NTDs) represents socioeconomic burden in most countries of Latin America. Chagas disease, a NTD, is caused by the parasite *Trypanosoma cruzi*. The disease can be mild, causing swelling and fever, or it can be long-lasting. Left untreated, it often causes heart failure. This study focused on *T. cruzi* lineages, emphasizing the gap of information from Central America and complementing what is known in North America. Our diverse collection of kissing bugs from North America (United States and Mexico) and Central America identified two of the major mitochondrial lineages circulating in these regions, both representing distinct clades within the already established three clusters of the *T. cruzi* parasite (mtTcI-mtTcIII): $mtTcI_{NA-CA}$ and $mtTcIII_{NA-CA}$. At a local scale, population genetic structure of *T. cruzi* revealed that genetic diversity has a notable geographic component. The important insights into the genetic and evolutionary diversity of *T. cruzi* in North and Central America provide not only the necessity for referencing genomes to identify lineages but the basis to develop more precise and comprehensive diagnostic assays to better detect *T. cruzi* infections.

## Introduction

Annually, parasitic infections cause more than one million deaths worldwide [1,2]. *Trypanosoma cruzi*, a hemoflagellated protozoan parasite and the causative agent of Chagas disease, estimated to be responsible for over 10,000 of these deaths although the actual mortality is unknown [1,3]. *Trypanosoma cruzi* comprises a morphologically cryptic group with genetically different lineages. The diverse outcomes of Chagas disease (i.e., cardiomyopathy, mega-colon and mega-esophagus) can be associated with distinct parasite lineages [4], which can also be traced back to their geography, ecology, virulence and transmission cycles (sylvatic and domestic) [5].

Currently, the *T. cruzi* clade is divided into six Discrete Typing Units (DTUs, [6]). These DTUs—designated as TcI through TcVI [7,8]—are based on nuclear DNA sequences from fragments of a single or only a few genes; a seventh clade is related to bats and called TcBat [9]. In contrast with the six nuclear DTUs, mitochondrial genes divide the *T. cruzi* clade into three discrete groups (mtTcI, mtTcII and mtTcIII), that somewhat reflect the nuclear DTUs [10]. Notably TcI corresponds to mtTcI, TcII to mtTcII and the hybrid lineages, TcIII-TcVI, fall within mtTcIII. From a phylogenetic perspective, differences between nuclear DTUs and mitochondrial clades are not a rare evolutionary occurrence. Tomasini and Diosque [11] suggested the hypothesis that mitochondrial introgression occurred between TcIII and TcIV along with nuclear hybridization between TcII and TcIII while others have proposed hypothesis such as the TcIII and TcIV are hybrids between TcI and TcII [12]. Although there is general agreement that hybridization plays a major role in the evolution of the group, the details represent an area of active research. The most current hypothesis is that TcI and TcII

represent ancestral lineages and TcIII–TcVI are hybrids; however, the evolutionary origins of the hybrids are currently unresolved [7,13,14]. Following the convention from the literature, we refer to the nuclear lineages as TcI–TcVI and the mitochondrial as mtTcI–mtTcIII. However, prior to 2016, mitochondrial lineages were not always recognized as such and/or were based on nuclear genes, with the result that some samples in GenBank and the literature are misclassified. Although the differences between nuclear and mitochondrial phylogenies has stimulated evolutionary hypotheses, what is lacking for a complete understanding of the evolution of *T. cruzi* is information about the lineages present in North and Central America.

The most comprehensive study of *T. cruzi* to date reported 90.7% of samples are from South America [15] highlighting the scarce sampling of Central America. A few studies include just a subset of countries in Central America, where TcI [16,17] and TcII [18] have been identified, and these results are based on 1–2 genes, and do not further classify "non-TcI" lineages. What little work has been done has identified TcI and TcIV in *T. dimidiata* s.l., the main vector of Chagas disease from Mexico to Colombia [19]. Therefore, it is unclear which nuclear or mitochondrial *T. cruzi* lineages are present in North and Central America, and how they relate to the better-known South American (SA) lineages.

The *T. cruzi* lineages present in North and Central America deserves to be explored further to not only understand the evolution and genetic variation of the *T. cruzi* group at both the regional and local scale, but also because this information is relevant to Chagas disease diagnostic tools. Accurate diagnosis of *T. cruzi* in patients with Chagas disease is based on serological tests that are supported using *T. cruzi* references strains as positive controls, currently these references strains are all from South America. These references and thus the diagnostic tests represent the genetic diversity of the geographic region where they were collected [20]. For the underrepresented regions of North and Central America, the lack of effective diagnostics may be due to the limited genetic studies of *T. cruzi* from these regions [21].

As *T. cruzi* data from North America became available, phylogenetic inference suggested continental genetic divisions [11]. Although this finding was consistent with studies that used a few different nuclear and mitochondrial molecular markers (i.e. SSU rDNA [9], *Dhfrs* [22]), so far, no study has addressed where such continental genetic divisions occur since critical sampling in Central America were lacking there.

In this study, we sequenced newly collected *T. cruzi* samples from North and Central America. Our genomic data was combined with publicly available reference genomes from South America. We examined two mitochondrial genes, NADH dehydrogenase subunit 1 (ND1) and cytochrome oxidase subunit II (COII), and expanded nuclear genomic sequence data from a few genes to include 1,563 conserved single copy genes across the genome using a reduced representation sequencing (RAD-seq) approach. Resolving the *T. cruzi* phylogeny emphasizes the importance of understanding the evolutionary history to address challenges such as the current lack of effective antiparasitic drugs to treat *T. cruzi* infection in humans and the importance of population genetics in drug development [23] and epidemiology [24,25].

In addition to continental-scale phylogeographic analysis, we used a population genetics approach to test for geographic patterns of genetic structuring within North and Central American *T. cruzi*. The combined phylogenetic and population genetic analyses to elucidate *T. cruzi* evolution is central to understanding disease transmission and developing effective treatments and diagnostic tools.

## Methodology

### Sample sites and *T. cruzi* from insect vectors

A total of 111 field-collected triatomines representing six species (*T. dimidiata*, *T. mopan*, *T. nitida*, *T. huehuetenanguensis*, *T. sanguisuga* and *T. recurva*) sampled across the United States and Mexico (NA) and Central America (CA) were sequenced for this study (Fig 1), and detailed information in S1 Table). The last three segments of the abdomen were used for DNA isolation with the Ezna Tissue DNA kit (Omega Bio-Tek, Georgia, GA, USA). We followed the manufacturer's tissue protocol for the first two steps, and the blood protocol for the remaining steps, with an additional incubation at 65˚C for 10 min followed by 95˚C for 5 minutes after the third step. Infection under field conditions (natural infection) prior to collection was evaluated by PCR using primers targeting the 18S (nuclear multi-copy 18s ribosomal subunit) and TcZ1, TcZ2 (nuclear multicopy satellite DNA) regions, following the PCR conditions previously described [26,27]. All PCR reactions were performed using a PTC-100 thermocycler (MJ Research, California, CA, USA). Electrophoresis of the amplified DNA used 1% agarose gels with 7.5 μg/mL of NANCY stain in TBE (90 mM Tris-borate, 1 mM EDTA, pH 8,0), followed by UV trans-illumination to observe the DNA bands.

### Sanger sequencing of COII and ND1 genes

For triatomine specimens where *T. cruzi* infection was detected by both nuclear gene assays, additional PCR reactions were used to amplify two *T. cruzi* mitochondrial genes: cytochrome oxidase subunit II (COII) and NADH dehydrogenase subunit 1 (ND1). Assay conditions and primer sequences for COII and ND1 are in Messenger [29]. PCR products of each gene were sequenced commercially (GeneWiz, Cambridge, MA, USA). Forward and reverse chromatograms were visually inspected using Sequencer version 5.3 [30] and confirmed as *T. cruzi* using NCBI-BLAST based on both maximum match to *T. cruzi* and an e-value of $<10^{-30}$ (the e-value represents the probability of a match by chance). Heteroplasmic mitochondrial genes or possible infection of a specimen by more than one *T. cruzi* genetic lineage or genetic strain were determined by the presence of double peaks.

The mitochondrial genes COII and ND1 have been widely sequenced previously for different *T. cruzi* strains over a broad geographic range, and many sequences are available in GenBank [31]. The reference sequences from GenBank used to perform phylogenetic analysis included: (1) samples representing all six nuclear genetic lineages and the three mitochondrial lineages, (2) isolates from both mammalian hosts and insect vectors, (3) samples sequenced for both mitochondrial genes and (4) as available, samples across the endemic region of each genetic lineage. With respect to (4) for genetic lineages with widely distributed data available (TcI and TcIV), samples across the whole distribution range were selected, whereas for genetic lineages with data restricted to South America (TcII, TcIII, TcV and TcVI), the geographic distribution was covered as much as possible (Fig 1, for further details S1 Table). In addition, *T. cruzi marinkellei* was included as an outgroup. In this study, we refer to broad genetic diversity (e.g., the nuclear based DTUs TcI-TcVI and mt TcI-TcIII) as genetic lineages and use the term genetic strain to refer to variation within lineages.

### Library preparation for the reduced genome representation: genotyping by sequencing (GBS)

Genomic DNA from the last three segments of the abdomen from naturally infected triatomine was sequenced using reduced representation Genotype By Sequencing (GBS) at Cornell University Genomics facility. GBS targets sites in low-copy genomic regions through the use

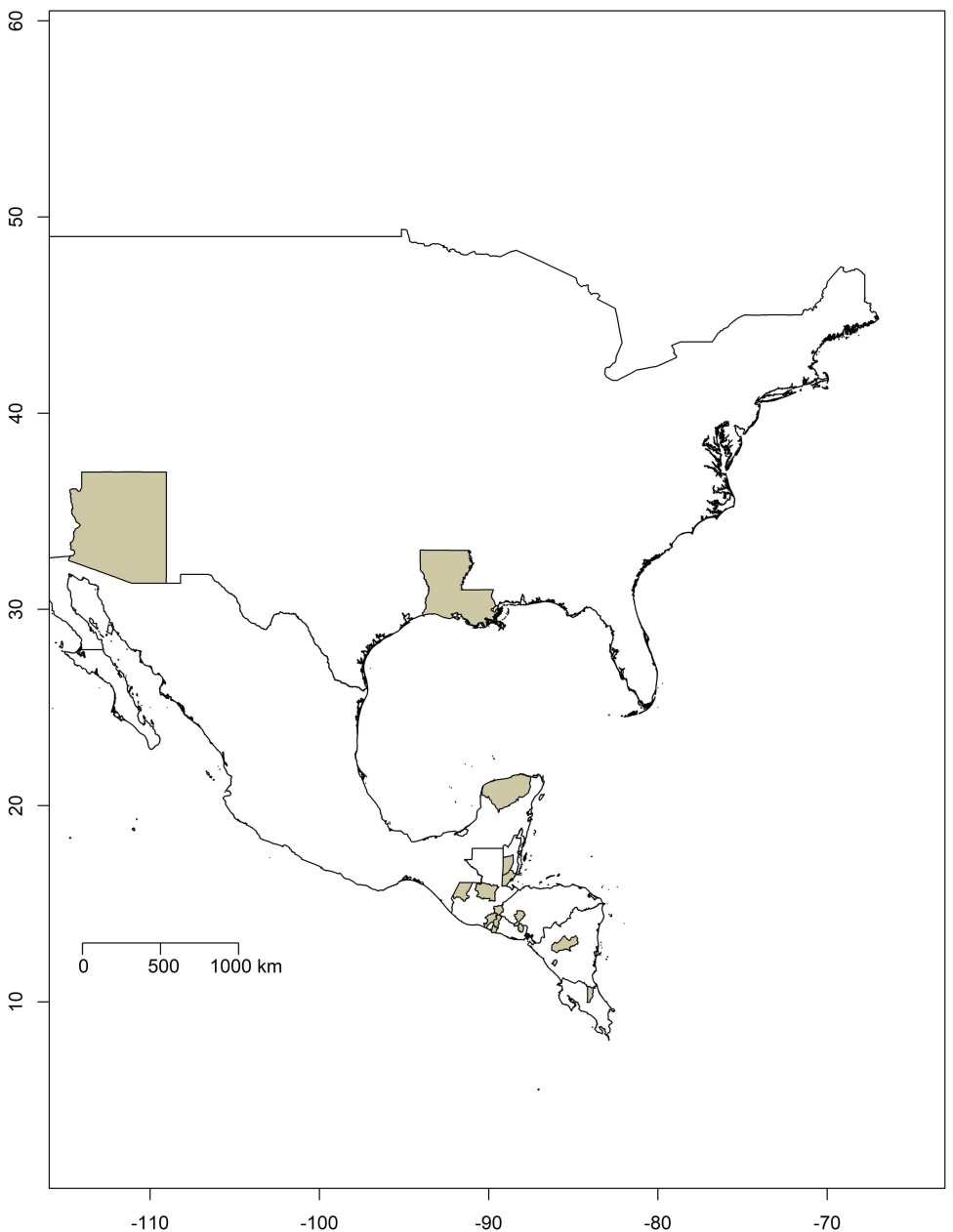

**Fig 1. Geographic distribution of field-collected triatomines sampled across North America (United States and Mexico) and Central America.** Sampled states in each country are shown in grey. Detailed information is in S1 Table. Base layer for each country and state/department were downloaded from the GADM database of Global Administrative Areas, version 3.6 (https://gadm.org/download_country.html) and maps were plotted using the R package mapdata [28].

of restriction enzyme digestion (also referred to as restriction site associated DNA sequencing, RAD-seq) [32]. From each triatomine specimen, a DNA library was constructed using the enzyme *PstI* with the recognition site: 5' CTGCA|G 3', 3' G|ACGTC 5'. Samples were run in a 48-plex genotyping array on an Illumina HiSeq Analyzer, producing 85 bp reads after trimming the 5-bp barcode and the fragment of the *PstI* cut site.

## Data analysis

**Strain typing and phylogenetic inference.** The mitochondrial DNA phylogeny of 111 samples of *T. cruzi* from naturally infected triatomines was determined based on the COII and ND1 genes. For each mitochondrial gene, we constructed a single matrix including our 111 samples, GenBank reference sequences and *Trypanosoma cruzi marinkellei* (outgroup) using Mesquite v3.04 [33]. Alignments by gene were performed using the Multiple Alignment using Fast Fourier Transform (MAFFT) algorithm [34] with the default parameters for Gap open and Gap extension penalties. Both ND1 and COII matrices were concatenated for phylogenetic analysis performed on MrBayes v3.2.6 [35] using the CIPRES Portal [36].

The nuclear DNA phylogeny of 27 samples of *T. cruzi* from naturally infected triatomines was determined based on SNPs from the GBS data. The difference between the nuclear and mitochondrial sample sizes reflects the higher DNA quality and quantity needed for genomic sequencing; only 27 out of the 111 samples "passed" the quality and quantity control. DNA quality was assessed by running 100 ng of each DNA sample on 1% agarose gels. Quantity control was assessed by an intercalating dye using the Qubit dsDNA HS assay following the manufactures' protocol, only samples with at least 30μL of genomic DNA at 50–100 ng/μl, were sent for sequencing.

Because genomic DNA was isolated from the last three segments of the abdomen from naturally infected triatomine, then GBS included sequences from the triatomine, *T. cruzi*, microbiome and blood meal sources. Therefore, read sequences were mapped to *T. cruzi* reference genomes to filter out all non-*T.cruzi* sequences. Because more than 50% of the *T. cruzi* genome consists of highly repetitive regions, mapping and SNP identification was restricted to the 1,563 CL Brener NonEsmeraldo-like conserved single copy nuclear genes as described in [37]. Mapping used Bowtie2 as in Reis-Cunha (2015) with the preset "very sensitive" setting and mismatch parameter = 1. After mapping, SNPs were retrieved using the pipelines ref_map.pl and *populations* from Stacks version 1.48 [38]. SNPs were called using a read depth (m) of 1 and including only loci present in at least 90% of the samples. Inter-lineage DTU analysis retrieved a total of 819 SNPs from 29 samples combined with 8 GenBank reference genomes whereas intra-lineage DTU analysis on only the TcI lineage retrieved 57 SNPs from 27 samples.

For both mitochondrial genes and nuclear SNPs, the best fitting nucleotide substitution models for the phylogenetic analysis were determined using JmodelTest with AIC [39]. Phylogenetic trees were constructed using MrBayes v3.2.6 [35] and the CIPRES Portal [36]. Support for phylogenetic trees was assessed by the bootstrap method using 1000 pseudo–replicates. The final phylogenetic tree images were built using FigTree v.1.4.2 software (http://tree.bio.ed.ac.uk/software/figtree/).

**Genetic diversity and haplotype networks.** Two estimators described genetic diversity at the population level: nucleotide and haplotype diversity. Nucleotide diversity, pi ($\pi$), is defined as the mean nucleotide differences between each pair of sequences, whereas haplotype diversity is defined as the chance that two randomly sampled alleles are different. Both estimators were calculated for mitochondrial sequence data with DnaSP version 6 [40]. DnaSP was also used to generate input files to calculate haplotype networks using the minimum spanning method in PopArt with the parameter epsilon = 0 [41].

**Demographic history at the regional scale.** Sequenced samples were classified by geographic region based on the collection location of the triatomines using biogeographic areas described previously [42]. Because of the limited numbers of samples for most lineages, Tajima's D [43] and Fu and Li's D [44] neutrality tests were used to evaluate the demographic history by region for each represented *T. cruzi* genetic lineage using DnaSP version 6 [40].

Both Tajima's D and Fu and Li's D, neutrality tests detect the effect of demographic changes on DNA sequence variations. Tajima's D measures the differences between the number of segregating sites and the average number of pairwise differences between each pair of haplotypes; D<0 indicates population expansion whereas D>0 supports a model of balancing selection. In contrast, Fu and Li's D test compares the number of derived singleton mutations and the average number of pairwise differences between each pair of haplotypes assuming an infinite sites (no recurrent mutation) model without recombination. As described by Fu [45], F is more sensitive to demographic expansion, usually showing negative values.

**Genetic structure of the Tc I lineage.** A Principal Component Analysis (PCA) followed by a Discriminant Analysis of Principal Component (DAPC) from Adegenet package version 2.1.3 in R [46,47] was used to evaluate genetic structure without an *a priori* grouping [48]. We use the a-score function to estimate the optimal number of PCs in the PCA step of DAPC. An a-score close to the maximum of 1 indicates that the DAPC solution is both strongly discriminating and stable, while low values (toward 0 or negative) indicate either weak discrimination or instability of the results. For the *T. cruzi* SNP data, the a-score analysis had a maximum value for five PCs explaining 79.86% of the total variance (S1 and S2 Figs). The DAPC maximized genetic differentiation based on these five PCs.

The optimal number of genetic clusters (*k*) was determined using the *find.clusters* function of the Adegenet package version 2.1.3 in R [46,47]. The *find.clusters* function identifies the optimal *k* using a k-means algorithm to find a partition of the data clustering objects based on similarity, ignoring any categorical label related to the object. To select the optimum *k* we used the Bayesian Information Criterion (BIC) where the optimal BIC is indicated by an elbow in the curve of BIC values as a function of *k*. Based on the BIC criterion our optimal value of *k* (number of genetic clusters) is also five (S2 Fig).

Using the PCA, a Neighbor-joining tree was also computed with the PC distances to estimate the genetic relationships among the inferred clusters. In order to evaluate clade support detected by the NJ tree, Nei's genetic distances for 57 SNPs among the 27 *T. cruzi* genomic samples were calculated and a UPGMA cladogram was inferred using the StAMPP function from stamppNeisD [49]. Clade support for the UPGMA cladogram was assessed by the bootstrap method using 1000 pseudo–replicates.

Finally, an Isolation By Distance (IBD) model based on a Mantel test was used to test whether genetic and geographic distances were correlated. Specifically, we tested if nearby individuals were more genetically similar than expected by chance, and if genetic differences increase linearly with geographic distances. The Mantel test was run using *mantel.randtest* function of the ade4 package, a dependency package of Adegenet.

## Results

Geographic sampling characterized *T. cruzi* mtTcI and mtTcIII lineages circulating in North and Central America and identified the most northern and first mitochondrial DNA sequence data for TcBat in Central America. Additionally, we found relative monophyly of *T. cruzi* lineages in North and Central America relative to South America. At the population level, the genome wide SNPs show genetic diversity associated with geographic distances in Guatemala and El Salvador. Details are below.

### *T. cruzi* lineages identified based on phylogeny

Phylogenetic analysis detected three mitochondrial genetic groups of *T. cruzi* from naturally infected triatomines collected across North, Central and South America. First, a mtTcI that corresponds to DTU TcI based on nuclear genes; second, a mtTcIII that corresponds to non-

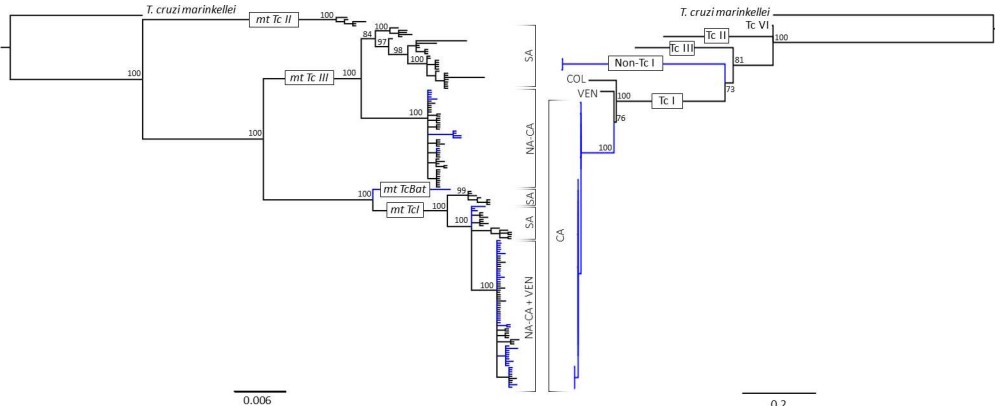

**Fig 2.** *Trypanosoma cruzi* **phylogenetic analysis based on mitochondrial DNA and nuclear single copy genes.** Left: Mitochondrial phylogeny based on COII-ND1 genes, inferred under the HKY model from 866 nucleotides from over 210 samples total (GenBank and study samples). Right: Phylogenomic analysis based on conserved nuclear single copy genes, inferred under the General Time Reversible model (GTR) from 819 SNPs from 35 samples total. Branch support is represented as percentages next to each clade. Blue branches represent *T. cruzi* sequences from this study; black branches represent *T. cruzi* GenBank sequences. GenBank accession numbers for the reference genomes used for the phylogenomic tree on the right: *T. marinkellei* (outgroup), Tc I (ADWP02 and AODP01), Tc II (ANOX01), Tc III (OGCJ01) and Tc VI (AAHK01). Abbreviations: North America (NA), Central America (CA), South America (SA), Venezuela (VEN), Colombia (COL).

TcI based on nuclear data (Fig 2) [we designate this as non-TcI because we did not have sufficient genetic information for further identification]; and third, a single mtTcBat sample with data only for the COII gene (Fig 2).

## North and Central America *T. cruzi* evolution relative to South America

Combining our data with previous data shows both mtTcI and mtTcIII lineages were detected over a wide geographic distribution, whereas the mitochondrial TcBat lineage was detected in a single triatomine (A2859) collected from a cave in Guatemala. A phylogeny based on the COII gene only clustered this sample (100% identity) with the only other TcBat COII sequence available, strain Tcc 1994, which was isolated from a bat from Sao Paulo, Brazil (KT337307.1) (S4 Fig).

All mtTcI samples also grouped with TcI based on their nuclear genetic sequences. Within mtTcI, all the samples from Central and North America were grouped in a monophyletic subclade along with three TcI sequences from Venezuela isolated from humans (JRcl4, EP strain and OPS21, refer to S5 Fig for further details), and one sample from Colombia from an opossum (VINC6 from *Didelphis marsupialis*) in this group. Nuclear SNPs revealed a TcI monophyletic subclade containing all the samples from Central America for which SNP data were available (i.e. Guatemala and El Salvador). GenBank strains from Venezuela and Colombia formed single branches, with the Venezuelan strain (JRcl4) closest to the Central American strains.

Similar to the mtTcI and TcI clades, mtTcIII (that corresponds to nuclear DTUs TcIII, TcIV, TcV and TcVI) was subdivided into two well-supported clades. One included all Gen-Bank samples from South America, the other included the new sequences fromNorth and Central America from this study as well as all of the GenBank sequences from North (100) and Central America (10) (Fig 1), with the exception of one sample from Guatemala that was isolated from a human and classified as TcIV based on nuclear genes (EU302217.1 Strain TcBRJ, refer to S5 Fig for further details) that fell within the South America clade. A sister

monophyletic clade of the nuclear TcI based on SNP data was well supported and conformed by two samples, NIC397 and FER530. Mitochondrial data was available only for one of these two samples, NIC397 which fell under the mtTcIII$_{NA-CA}$ clade based on mitochondrial data.

## Evolutionary history for mitochondrial *T. cruzi* lineages across the Americas

The haplotype network for the mtTcI lineage shows geographic structure separating South America from North and Central America (Fig 3), except for the three TcI sequences from Venezuela isolated from humans and one sample from Colombia isolated from opossum that clustered with North and Central America in the mtDNA phylogeny. North and Central America mtTcI haplotypes are separated by three differences from the South America haplotypes. There was higher haplotype diversity in the South America region (H > 0.9), compared to North and Central America, 0.2 and 0.4 respectively (Table 1).

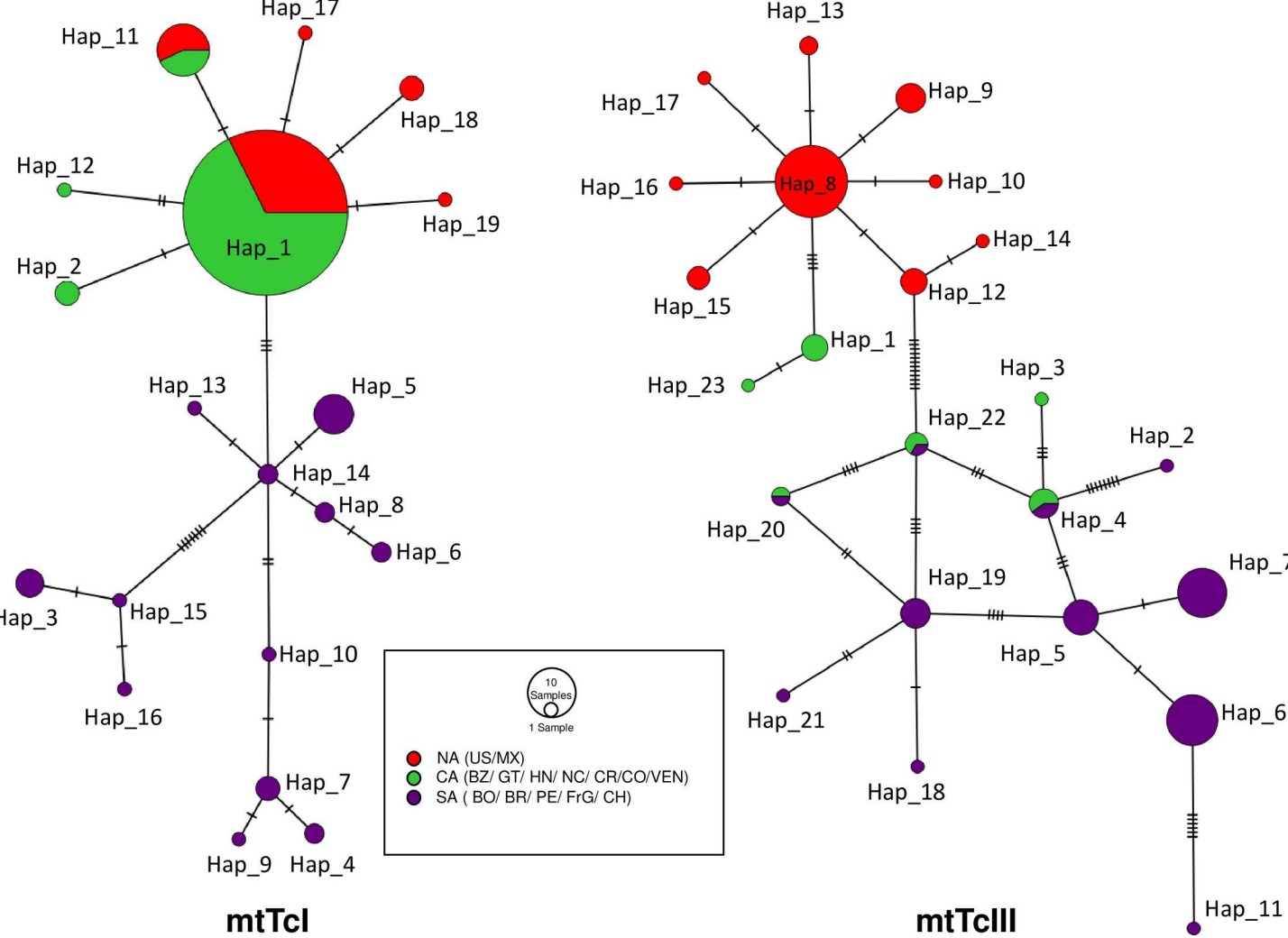

**Fig 3. Minimum Spanning Haplotype Networks of mtTcI and mtTcIII lineages. Left: Haplotype Network for the mtTcI lineage.** Right: Haplotype Network for the mtTcIII lineage. For both haplotypes networks, NCBI references and our samples were included (mtTcI N = 190 and mtTcIII N = 109). Circle size is proportional to the number of samples for the haplotype present. Abbreviations: US: United States, MX: Mexico, BZ: Belize, GT: Guatemala, HN: Honduras, NC: Nicaragua: CR: Costa Rica, CO: Colombia, VZ: Venezuela, BO: Bolivia, BR: Brazil, PE: Perú, FrG: French Guiana and CH: Chile.

**Table 1. Genetic diversity and neutrality tests for mtTcI clade by region.**

| Statistics | North America | Central America | South America |
|---|---|---|---|
| Sample size | 58 | 99 | 33 |
| H | 0.409 | 0.207 | 0.924 |
| pi (π) | 0.0008 | 0.0004 | 0.008 |
| S | 6 | 5 | 27 |
| **Tajima's D** | -1.432 | -1.61 | -0.425 |
| **Fu and Li's D** | -1.633 | -2.14 | -0.286 |

The haplotype network for the mtTcIII lineage shows a more evident continental genetic division (Fig 3), where Central America mtTcIII haplotypes are separated by three differences from North America haplotypes. Whereas North America mtTcIII haplotypes are separated by 14 differences from South America. Thus, Central America and South America are separated by more than 15 differences, except again for the one sample from a human from Guatemala (EU302217.1 Strain TcBRJ), appearing within the South America subclade.

All three regions showed high and similar haplotype diversity (H) that ranged from 0.80 to 0.84 (Table 2). Both neutrality tests indicated significantly negative D only for the mtTcIII lineage from Central America. The difference between H and nucleotide diversity (π) suggests that populations might have experienced a bottleneck followed by population expansion in the Central America region.

## Genetic Structure for the nuclear Tc1 lineage at a local scale

Samples typed as TcI by the nuclear SNPs from reduced genome representation sequencing showed genetic structure when evaluated at a local scale (Fig 4). Four of the locations are connected geographically (Jutiapa, Chiquimula, Santa Ana and Sonsonate); specifically, the extreme west and east locations, Huehuetenango and San Fernando, are about 380 and 230 km away from the Jutiapa-Santa Ana-Sonsonate, Chiquimula locations. The Discriminant Analysis of Principal Component Analysis (Fig 4A) along with the Neighbor-joining tree inferred from the PCA distances (NJ, Fig 4B) supported five clusters where all locations were present in at least two clusters.

The UPGMA cladogram supported the same five genetic clusters (Fig 5) previously defined by the NJ tree, however only clusters 3 and 4, along with one sub-group within cluster 1 was highly supported by our bootstrap analysis. The presence of long branches between each cluster reflects the high number of substitutions among them. Although all locations were represented in at least two clusters, cluster 2 and 4 include samples from the closest locations Jutiapa-Santa Ana-Sonsonate, whereas Cluster 3 and the sub-group within cluster 1 includes samples only from Huehuetenango.

**Table 2. Genetic diversity and neutrality tests for mtTcIII lineage by region.**

| Statistics | North America | Central America | South America |
|---|---|---|---|
| Sample size | 48 | 6 | 55 |
| H | 0.827 | 0.800 | 0.840 |
| pi (π) | 0.0019 | 0.009 | 0.0065 |
| S | 11 | 23 | 25 |
| **Tajima's D** | -1.385 | -1.500* | -1.150 |
| **Fu and Li's D** | -1.40 | -1.539* | -2.381 |

* $p < 0.05$

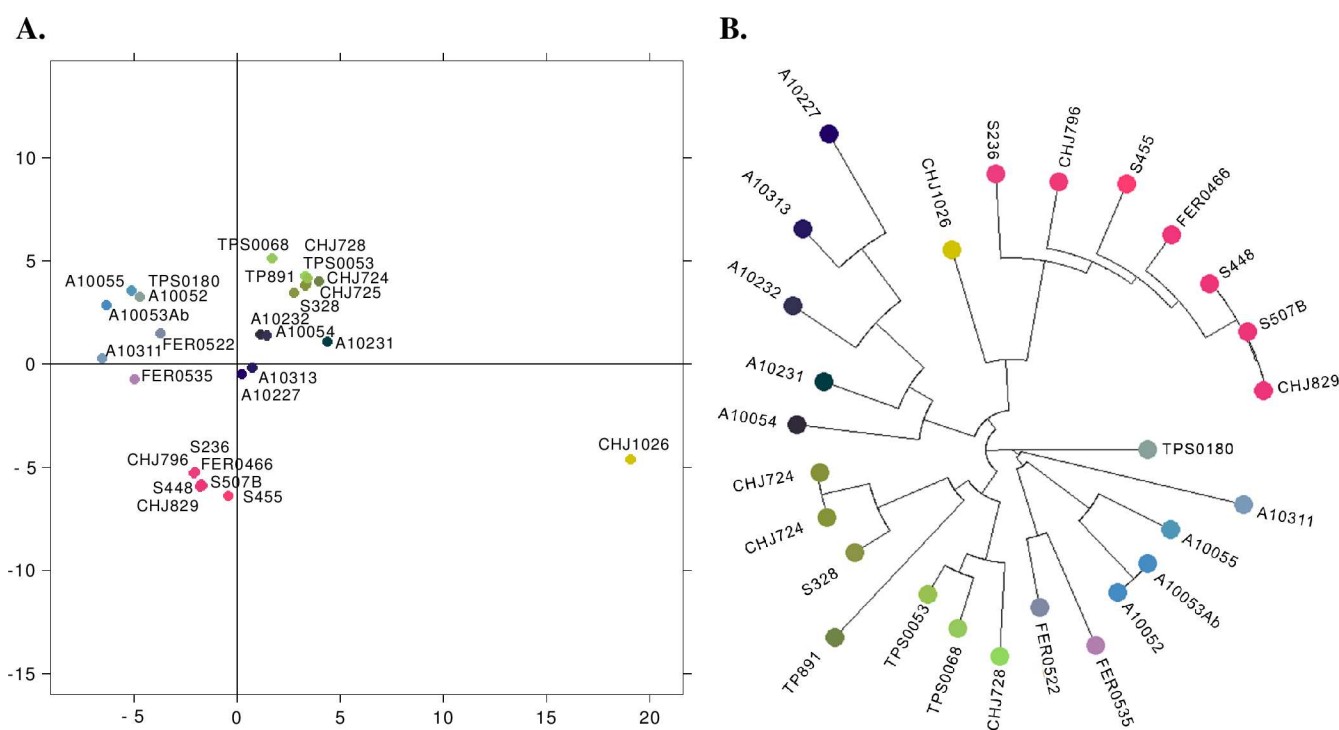

**Fig 4. Discriminant Analysis of Principal component analysis (DAPCA) and Neighbor Joining Tree for the nuclear TcI lineage from Guatemala and El Salvador.** A) Discriminant Analysis of Principal Component, B) Neighbor-Joining Tree inferred from the PCA distances.

**Isolation by Distance (IBD):** The IBD test supported the hypothesis that nearby individuals are genetically more similar than expected by chance, i.e., there is a significant relationship between genetic and geographic distances. Because of the strong difference of the Huehuetenango cluster (cluster 3, and sub-cluster within cluster 1, Fig 5) we repeated the analysis omitting this cluster and still found significant IBD. The Mantel test slope for the estimate is equal to 0.407 (p<0.001) for the scatterplot of genetic and geographic distances (Fig 6).

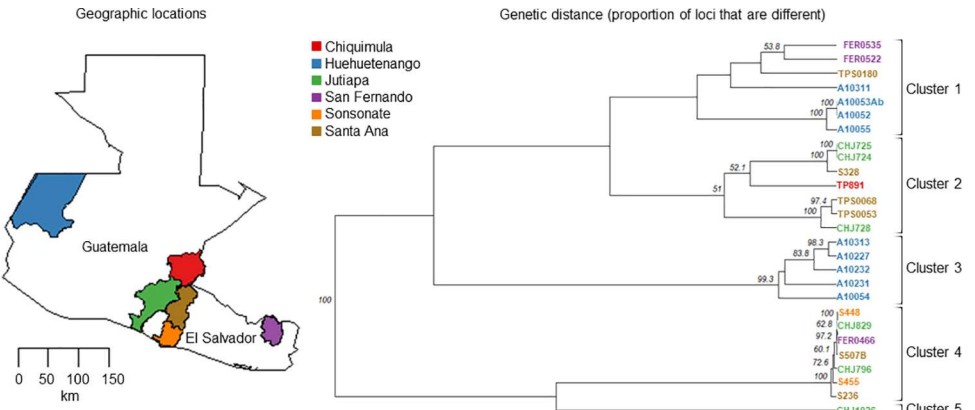

**Fig 5. UPGMA Dendrogram based on Nei's genetic distances among TcI samples across locations from Guatemala and El Salvador.** Node support was assessed by bootstrap. Nei's genetic distances matrix involved 57 SNPs from 27 samples. Colors represent each of the geographic locations evaluated. Base layer for each country and state/department were downloaded from the GADM database of Global Administrative Areas, version 3.6 (https://gadm.org/download_country.html) and maps were plotted using the R package mapdata [28].

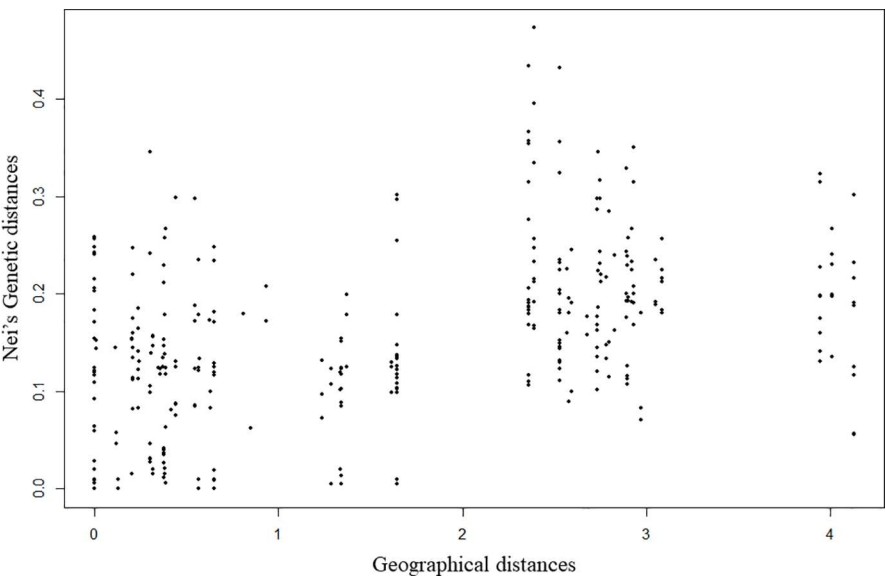

**Fig 6. Correlation of individual Nei's genetic and geographical distances.**

## Discussion

The extensive geographic sampling of *T. cruzi* from North and Central America combined with the publicly available South American lineages give robust support to the continental genetic divisions across the Chagas endemic area. Our results suggest there is limited movement across geographic boundaries because they show that both the mtTcI and nuclear TcI indicate a single monophyletic clade with intra-lineage genetic structure reflecting geography. Specifically, for the mitochondrial data, all our samples from North and Central America clustered into a single monophyletic subclade along with three mtTcI sequences from Venezuela isolated from humans, which are likely the most mobile of the *T. cruzi* mammal hosts and vectors. Interestingly, the nuclear SNP data showed similar results, all TcI samples from Central America (i.e., Guatemala and El Salvador) grouped into a monophyletic subclade that included a Venezuela sample (JRcl4) as the closest relative. As mentioned above, JRcl4 was collected from a human patient. Thus, our mitochondrial and nuclear results combined not only suggest that strains from Venezuela are the closest relatives to Central American strains with perhaps some movement, but also support the hypothesis that North and Central America evolved independently from South American lineages. This is particularly interesting because TcI is the most diverse lineage with the widest geographic distribution [50].

The mitochondrial genetic data from COII-ND1 genes from *T. cruzi* lineages circulating in North and Central America supported the three major mitochondrial clades shown by [51] and [10]. The monophyly of *T. cruzi* in N and Central America relative to South America revealed a new sister clade to mtTcIII also supporting limited movement across geographic boundaries for this group. This clade included samples from North and Central America, thus we referred to it as mtTcIII$_{NA-CA}$. Our analysis with more samples indicates that GenBank sequences from samples previously identified as TcIV$_{NA}$ fell within our mtTcIII$_{NA-CA}$ lineage [22] in contrast to a previous study where those GenBank sequences identified as TcIV$_{NA}$ were not resolved with the COII and CytB genes [10]. Thus, our findings suggests that mtTcIII from North and Central America evolved independently from South America has limited movement across geographic boundaries. In addition to the comparison with South America, intra-lineage genetic structure was evident within North and Central America for the mtTcIII clade

(Fig 3). Both neutrality tests (Tajima's D and Fu and Li's D), suggest that mtTcIII clade deviates from the neutral model. In addition to this, all three regions showed high (> 0.8) but similar values of haplotype diversity with haplotypes being different among North, Central and South America again reflecting the different evolutionary paths of these groups.

Isolation by distance with migration between Guatemala and El Salvador was shown for TcI within the lineage and at the population level. Previously, genotypes of this lineage were reported to be associated with particular triatomine species [52], ecotopes [53,54,55] and biomes [50]. Although we did not assess those variables, our data showed genetic structure associated with geography, in agreement with [56]. Nearby individuals tend to be genetically more similar than expected by chance, and genetic differences increase with geographic distances. Although there is genetic structure related to geography, there was also evidence of some mixture among locations, thus our results not only suggest movement among closer locations such as Jutiapa, Santa Ana and Sonsonate (i.e., clusters 2 and 4) but also could suggest incomplete lineage sorting. Movement could reflect movement of the mammal hosts, rather than by the vector (*T. dimidiata*) because mammals likely move more than the reported 40-60m movement of *T. dimidiata* in a 2-week period [57]. In the other hand, incomplete lineage sorting is possible because these locations retain several, old distinct lineages with rare sexual recombination, thus staying distinctive.

## Conclusion

These results combined with the identification of the distinct genetic lineages of *T. cruzi* circulating across the North and Central America are important to understand the ecology of disease transmission and the clinical outcome of Chagas disease. The identification of genetically distinct lineages of *T. cruzi* in North and Central America relative to South America supports that the diversification occurred after the separation of North and Central America from South America. These results also compare to the triatomine species in the region, *T. dimidiata*, which showed a distribution pattern associated with geology at the regional scale, but also associated with ecology at local scale [58].

The distinct genetic lineages in North and Central America could explain why the sensitivity of current diagnostic tests, derived almost exclusively from South America samples are less sensitive in Central America [21]. A recent study suggested that such lack of sensitivity of current diagnostic tests relates to the antigenic diversity in *T. cruzi* [56]. This highlights the need to have reference genomes and diagnostic reagents that recognize North and Central American *T. cruzi* strains. The work reported here demonstrates that the genetic population structure of TcI, the most prevalent lineage circulating in Guatemala and El Salvador, includes significant Isolation By Distance as well as evidence of migration among local villages. The need of triatomines naturally infected with *T. cruzi* from the regions that did not have representation on our population genetic structure analysis is crucial as this can help determine the main dispersal mechanisms that are operating; possibilities include vector and mammal movement. The use of remote sensing technology as a complementary tool, can help us elucidate what environmental factors are related to vector-borne disease maintenance and transmission, a field known as landscape epidemiology [59]. Additionally, the observation of distinct lineages of *T. cruzi* can have implications not only in the development of effective diagnostics but also for pathogenicity and infectivity, all areas that need further research. Sensitive and specific diagnostic tools require clear identification of the lineages and their evolutionary history as well as understanding small scale genetic diversity to characterize how the parasite changes in space and time. The integration of vector ecology along with genetics studies on the vector and pathogen are becoming crucial for disease epidemiology.

Additionally, our mitochondrial data confirm the first record of TcBat in Guatemala and the most northern record for TcBat. Based on comparison with sequences in Genbank from Brazil and Colombia, we identified TcBat in a *T. dimidiata* triatomine collected in Alta Verapaz, Guatemala. Previously, TcBat had only been reported in Brazil [9] Panama [60] and Colombia [61], with both nuclear and mitochondrial genetic data available for Brazil and Colombia and only nuclear for Panama.

## Supporting information

**S1 Fig. PCA Eigenvalues.**
(TIF)

**S2 Fig. Plot of the BIC values for the nuclear Tc1 lineage dataset from Guatemala and El Salvador.**
(TIFF)

**S3 Fig. a-Score analysis.**
(TIF)

**S4 Fig. *Trypanosoma cruzi* phylogenetic analysis based on mitochondrial DNA.** Mitochondrial phylogeny based on COII gene, inferred under the GTR model from 513 nucleotides from reference and newly sequenced samples.
(TIF)

**S5 Fig. *Trypanosoma cruzi* phylogenetic analysis based on mitochondrial DNA.** Mitochondrial Phylogeny based on COII-ND1 genes, inferred under the HKY model from 866 nucleotides from over 210 samples total (reference and newly sequenced samples).
(TIF)

**S1 Table. GenBank accession numbers for two mitochondrial genes NADH dehydrogenase subunit 1 (ND1) and cytochrome oxidase subunit II (COII) examined organized by Sample ID, country and triatomines species.** Nuclear DTUs are based on [8] consensus intraspecific nomenclature for *T. cruzi* and mitochondrial nomenclature is based on [10].
(XLSX)

**S2 Table. Reference GenBank accession numbers for two mitochondrial genes NADH dehydrogenase subunit 1 (ND1) and cytochrome oxidase subunit II (COII) examined organized by Sample ID, country and triatomines species.** Nuclear DTUs are based on [8] consensus intraspecific nomenclature for *T. cruzi* and mitochondrial nomenclature is based on [10].
(XLSX)

## Acknowledgments

This article was drafted and reviewed during the phylogenetic systematics course at University of Vermont. The authors thank Ingi Agnarsson for guidance in the scientific development of the manuscript and for his valuable comments. We thanks to Norman Beatty for providing additional kissing bugs from Arizona and Lucía Orantes and Elizabeth Solórzano for processing part of the bugs for DNA extraction and library preparation for the reduced genome representation sequencing. The authors also thank Stephen Keller for valuable input to improve the manuscript. The material published reflects the views of the authors and should not be misconstrued to represent those of the Department of the Army, the Department of Defense, USDA, NSF or other funding bodies.

## Author Contributions

**Conceptualization:** Raquel Asunción Lima-Cordón, Sara Helms Cahan, Patricia L. Dorn, Silvia Andrade Justi, Lori Stevens.

**Data curation:** Raquel Asunción Lima-Cordón.

**Formal analysis:** Raquel Asunción Lima-Cordón, Sara Helms Cahan, Cai McCann, Patricia L. Dorn, Lori Stevens.

**Funding acquisition:** Raquel Asunción Lima-Cordón, Sara Helms Cahan, Patricia L. Dorn, Antonieta Rodas, María Carlota Monroy, Lori Stevens.

**Investigation:** Raquel Asunción Lima-Cordón, Cai McCann.

**Methodology:** Raquel Asunción Lima-Cordón.

**Project administration:** Patricia L. Dorn, Antonieta Rodas, María Carlota Monroy, Lori Stevens.

**Resources:** Sara Helms Cahan, Patricia L. Dorn, Antonieta Rodas, María Carlota Monroy, Lori Stevens.

**Supervision:** Lori Stevens.

**Validation:** Raquel Asunción Lima-Cordón, Cai McCann.

**Visualization:** Raquel Asunción Lima-Cordón.

**Writing – original draft:** Raquel Asunción Lima-Cordón, Lori Stevens.

**Writing – review & editing:** Raquel Asunción Lima-Cordón, Sara Helms Cahan, Cai McCann, Patricia L. Dorn, Silvia Andrade Justi, Antonieta Rodas, María Carlota Monroy, Lori Stevens.

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
