## [Decision Letter · Decision Letter 0]

9 Sep 2021

Dear Ms Lima-Cordón,

Thank you very much for submitting your manuscript "Insights from a comprehensive study of Trypanosoma cruzi: a new mitochondrial clade restricted to North and Central America and genetic structure of TcI in the region." for consideration at PLOS Neglected Tropical Diseases. As with all papers reviewed by the journal, your manuscript was reviewed by members of the editorial board and by several independent reviewers. The reviewers appreciated the attention to an important topic. Based on the reviews, we are likely to accept this manuscript for publication, providing that you modify the manuscript according to the review recommendations. 

Sincerely,

Walderez O. Dutra, PhD.

Deputy Editor

Ana Rodriguez

Deputy Editor

Reviewer's Responses to Questions

**Key Review Criteria Required for Acceptance?**

**Methods**

-Are the objectives of the study clearly articulated with a clear testable hypothesis stated?

-Is the study design appropriate to address the stated objectives?

-Is the population clearly described and appropriate for the hypothesis being tested?

-Is the sample size sufficient to ensure adequate power to address the hypothesis being tested?

-Were correct statistical analysis used to support conclusions?

-Are there concerns about ethical or regulatory requirements being met?

Reviewer #1: Objectives are clearly stated. Study design is adequate. 

Population needs some clarification as mentioned in the comments. 

Sample size is sufficient but limitations need to be added to discussion.

No ethical concerns.

Reviewer #2: (No Response)

**Results**

-Does the analysis presented match the analysis plan?

-Are the results clearly and completely presented?

-Are the figures (Tables, Images) of sufficient quality for clarity?

Reviewer #1: Figures 1 needs revamping. Some mislabeling is found in the manuscript. 

Quality of figures needs assistance. Unclear if related to submission system. 

Results need some clarification as mentioned in the comments below.

Reviewer #2: (No Response)

**Conclusions**

-Are the conclusions supported by the data presented?

-Are the limitations of analysis clearly described?

-Do the authors discuss how these data can be helpful to advance our understanding of the topic under study?

-Is public health relevance addressed?

Reviewer #1: Need limitations to be discussed.

Reviewer #2: (No Response)

**Editorial and Data Presentation Modifications?**

Reviewer #1: (No Response)

Reviewer #2: (No Response)

**Summary and General Comments**

Reviewer #1: Suggestions for authors:

Author summary:

Line 54 needs reference. Fix: "Latinamerica" to Latin America

Line 58: "Wide collection" is a valid point when discussing the countries your specimens were collected in but I think it is better stated as "diverse collection of kissing bugs from United States, Mexico and Central America..." For example, in Mexico alone there are several regional differences that have been described with different populations of triatomines and T. cruzi linkage. Looking at your samples from Mexico, they were collected in the Yucatan but lack the majority of the country. I think it is important that the readers understand this from the beginning. 

Introduction:

Line 70: even though "10,000 deaths" are often cited for annual mortality, this is likely grossly inaccurate and several experts acknowledge this fact. I think it is important that we recognize this and state this an estimation but mortality is truly unknown. 

Line 72: I agree that certain lineages are associated with distinct patterns of clinical disease but this is not always the case. For example we do see gastrointestinal disease in those who were likely infected in regions of TcI predominant regions. I think it is better said that "can be associated with distinct parasite lineages..."

Methodology:

Line 139-141: Will need more details pertaining to sample sites. What regions of the U.S., Mexico, and Central American countries (states). Figure 1 needs revamping and is an important visual element of the study. I do not think it is useful as it is being presented. I suggest adding GIS mapping for species collected based off region as it pertains to number sequenced and data from GeneBank. Some regions had more sequencing than others. 

Why was there a section for "Genetic structure of the Tc I lineage within Central America" and not one for describing North America? Need further clarification. 

Results:

Line 155 and Line 345: I am confused with two figure 1. Are we missing a figure? I think authors met Fig 3?

Line 297 and Line 376: I think we have similar issue. Line 376 is Fig 4.

Line 398: Fig 6?

Discussion:

Need clearly defined limitations of the study discussed in this section. There are several limitations, which includes sample size. One strength is the data presented from samples collected in Guatemala and El Salvador, but also a limitation as other sampled regions did not have the same variables for analysis. 

I agree with the findings of the author and appreciate that they used the terminology "suggest" and their findings "support" regional evolutionary patterns for T. cruzi lineage. Something not discussed was Mexico and the distinct lineages found in this diverse region that is endemic to Chagas disease. This needs to be considered in the discussion as well as including some data presented by other researchers, such as work done by Herrera and Dumonteil. In the United States, how did your findings correlate with similar investigations looking at phylogeny of T. cruzi strains? This is why I think it is important to include which states those samples were collected in. The U.S. has a very diverse population of triatomines among a large area of geography.

Conclusion:

Line 448: Consistency with the way we describe our Chagas regions. I suggest, "North America, Mexico, and Central America". Triatomines are increasingly being found in regions we did not think they existed. Like certain Caribbean islands and possibly Canada someday with global warming and environmental changes.

Line 450 - 454: I agree the findings support the idea of regional diversification and separation of NA and CA from SA lineage. Again, I go back to Mexico, as T. cruzi in Mexico specifically has likely a similar regional diversification pattern. I would also suggest this is likely in the United States as well. This is why I having a hard time putting Mexico under the umbrella of "North America". I would suggest separating these two regions for this paper. 

Interesting findings are reported with TcBat, which will add to the literature on new region of isolation in Guatemala. 

General recommendations: 

When discussing North America and Central America, will need to be consistent throughout manuscript in regards to using abbreviations. Also need to be consistent with order at which you are discussing the regions studied. In regards to Mexico, I am not sure how you are going to present this data. In some instances Mexico is considered North America, and in others it is not. When discussing North America with regards to the western hemisphere, it also includes countries in Central America and the Caribbean islands. This can be tricky when reporting data for Chagas disease. My suggestion is be consistent throughout manuscript. North America, Mexico, and Central America is likely the best approach. 

I would also be consistent when describing the vector. I would suggest using triatomine throughout. 

Please ensure all the names of the authors and those found in the acknowledgements are spelled correctly.

Reviewer #2: (No Response)

PLOS authors have the option to publish the peer review history of their article (what does this mean?). If published, this will include your full peer review and any attached files.

Reviewer #1: No

Reviewer #2: No

Figure Files:

Data Requirements:

Reproducibility:

References

---

## [Decision Letter · Decision Letter 1]

2 Dec 2021

Dear Ms Lima-Cordón,

We are pleased to inform you that your manuscript 'Insights from a comprehensive study of Trypanosoma cruzi: a new mitochondrial clade restricted to North and Central America and genetic structure of TcI in the region.' has been provisionally accepted for publication in PLOS Neglected Tropical Diseases.

Best regards,

Walderez O. Dutra, PhD.

Deputy Editor

Ana Rodriguez

Deputy Editor

Reviewer's Responses to Questions

**Key Review Criteria Required for Acceptance?**

**Methods**

-Are the objectives of the study clearly articulated with a clear testable hypothesis stated?

-Is the study design appropriate to address the stated objectives?

-Is the population clearly described and appropriate for the hypothesis being tested?

-Is the sample size sufficient to ensure adequate power to address the hypothesis being tested?

-Were correct statistical analysis used to support conclusions?

-Are there concerns about ethical or regulatory requirements being met?

Reviewer #1: Objectives of the study are met and authors have addressed comments left from the two reviewers. Population of studied triatomines is more clearly defined in revised manuscript.

**Results**

-Does the analysis presented match the analysis plan?

-Are the results clearly and completely presented?

-Are the figures (Tables, Images) of sufficient quality for clarity?

Reviewer #1: Results are clearly understood and revisions have been made based off comments from reviewers. Figures and tables have been revised and improved.

**Conclusions**

-Are the conclusions supported by the data presented?

-Are the limitations of analysis clearly described?

-Do the authors discuss how these data can be helpful to advance our understanding of the topic under study?

-Is public health relevance addressed?

Reviewer #1: Conclusions are just and presented clearly. Data is discussed with public health relevance being addressed. Limitations have been added to revised manuscript.

**Editorial and Data Presentation Modifications?**

Reviewer #1: I recommend accepting the revised manuscript. Some grammatical errors still exist but can be addressed at the editorial stage.

**Summary and General Comments**

Reviewer #1: Overall the study conducted is revealing new insights into the phylogentics of T. cruzi in North (US and Mexico) and Central America. Comments from reviewers have been addressed and revised manuscript is ready for publication. Thank you for your work in the field of Chagas disease and Trypanosoma cruzi research.

PLOS authors have the option to publish the peer review history of their article (what does this mean?). If published, this will include your full peer review and any attached files.

Reviewer #1: **Yes: **Norman L. Beatty, MD, University of Florida College of Medicine, Gainesville, FL, USA

---

## [Editor Report · Acceptance letter]

9 Dec 2021

Dear Ms Lima-Cordón,

We are delighted to inform you that your manuscript, "Insights from a comprehensive study of Trypanosoma cruzi: a new mitochondrial clade restricted to North and Central America and genetic structure of TcI in the region.," has been formally accepted for publication in PLOS Neglected Tropical Diseases.

Best regards,

Shaden Kamhawi

co-Editor-in-Chief

Paul Brindley

co-Editor-in-Chief
